# Optimization of a Reference Kinetic Model for Solid Oxide Fuel Cells

**Fiammetta Rita Bianchi [1], Barbara Bosio [1,\*], Arianna Baldinelli [2], and Linda Barelli [2]**

[1] PERT-Department of Civil, Chemical and Environmental Engineering, University of Genoa, Via Opera Pia 15b, 16145 Genoa, Italy; fiammettarita.bianchi@edu.unige.it (F.R.B.)

[2] Department of Engineering, University of Perugia, Via Duranti 93, 06125 Perugia, Italy; arianna.baldinelli@unipg.it (A.B.); linda.barelli@unipg.it (L.B.)

**\*** Correspondence: barbara.bosio@unige.it (B.B.); Tel.: +39-010-3356505

**Abstract:** Solid oxide fuel cells (SOFCs) stand out among other fuel cell types because of their specific characteristics. The high operating temperature permits to reach optimal conductivity and favours kinetics without requiring noble metal catalysts. The SOFC behaviour analysis is fundamental to optimise operating conditions and to obtain the best performance. For this purpose, specific models are studied to investigate the electrochemical kinetics, which is the most critical aspect in the simulation. This is closely linked to cell materials and structure, as well as to operating conditions (feed composition and temperature above all) that influence cell polarization effects. The present work aims at evaluating these contributions by means of a semi-empirical kinetic formulation based on both theoretical and experimental approaches. A dedicated experimental campaign on an anode-supported NiYSZ/8YSZ/GDC-LSCF button cell is performed to identify experimental parameters. Each working variable is changed singularly to understand its specific effect, avoiding the overlap of multiple effects. The studied kinetics is validated using a 0D model to evaluate global cell operation, and a 1D model to estimate occurring mechanisms along anode thickness. The comparison between experimental and simulated data allows a preliminary validation of the proposed model, providing a base for subsequent more specific studies.

**Keywords:** anode-supported solid oxide cell; 0D electrochemical kinetic model; button cell experimentation; reacting composition profile

## 1. Introduction

The remarkable increase of atmospheric greenhouse gas level and consequent climate changes have forced the development of alternative more sustainable energy sources. Among these, fuel cells (FCs) are promising devices as they can continuously generate electricity without dangerous emissions, obtaining water as the main product. Their high efficiency is owing to the direct conversion from chemical to electric energy. Among different FC types, solid oxide fuel cells (SOFCs) are successfully used as stationary heat and power plant, thanks to their better performance owing to the high operating temperature. In anionic-conductive electrolyte SOFCs, oxygen is reduced at cathode and generates $O^{2-}$ ions (Equation (1)), which migrate through the electrolyte to the anode, where they oxidize hydrogen, producing electric current and water (Equation (2)). The resulting global reaction is expressed by Equation (3).

$$\frac{1}{2}O_2 + 2e^- \rightarrow O^{2-} \tag{1}$$

$$H_2 + O^{2-} \rightarrow H_2O + 2e^- \tag{2}$$

$$H_2 + \frac{1}{2}O_2 \rightarrow H_2O \tag{3}$$

This electrochemical process allows a better reaction control, for each single atom, in comparison with chemical combustion. FCs are quite flexible devices and the obtained performance is independent from plant size. Moreover, noise and vibrations are minimum, as no mobile parts are present. However, a further technological improvement is necessary to make FCs competitive on market.

For this purpose, the system simulation is fundamental to optimise SOFC design and operating conditions. Different models are proposed to study the electrochemical kinetics and to estimate different cell polarization effects. They are usually based on semi-empirical correlations, which request the comparison with experimental data for the detection and the validation of parameters [1–3]. Yet, a theoretical approach can also be used [4]. Material, thermal, and momentum balances are introduced in the simulation to have a better system description. A large number of models have been developed, which range from 0D to 3D. In the first case, the cell is reduced to a point to evaluate its global performance [5]. Meanwhile, the 3D simulation studies the cell local behavior; the distribution of chemical-physical properties along three spatial coordinates are obtained [6,7]. 1D or 2D models are a compromise between these two situations; in these cases, the analysis only focuses on the main system axes. In 1D simulations, balances are developed along the flow direction [8], whereas in 2D approaches, the control volume can be the cell plane [9] or the cross-section [10]. The choice of each model depends on the final goal and application. A 0D approach requests minor computational effort, so it is suitable for the whole power plant simulation. Meanwhile, when the attention focuses on how operating conditions, external factors, and degradation influence cell materials and local characteristics, a model with higher level of precision describes better FCs.

The main difficulty in SOFC simulation is the specific kinetics identification, to evaluate the different polarizations, which penalize cell voltage: ohmic, activation, and concentration overpotentials. Several approaches are proposed in the literature. The ohmic overpotential is directly proportional to current density and represents the cell resistance at charge transport. It usually depends on system geometry and material conductivity [1,5,11,12]. The activation overpotential, related to electrochemical reactions, is the loss that occurs at three phase boundary (TPB). The charge transfer step between electronic and ionic conductors requests an extra-potential to overcome the energy barrier and, therefore, to proceed at the desired rate. To solve the Butler–Volmer equation, three different formulations are proposed: the linear, exponential, and hyperbolic sine one. Nonetheless, the first two are effective only at low and high overpotential, respectively [13]. Thereby, the hyperbolic sine equation is commonly used, assuring a wider validity range [5,10,14]. The activation polarization contribution is a function of the rate between effective and equilibrium current density. According to the literature, this last term is expressed through several operating parameters. The equilibrium current density can be a constant value obtained by experimental data fitting [14], depending on temperature [5,10] and components composition [1,15]. More specific approaches introduce TPB length to underline electrocatalyst performance [11] or to evaluate composition dependence, considering the rate-limiting step of electrochemical reactions [16]. Finally, the concentration overpotential takes into account resistances owing to diffusion mechanisms. This term depends on the limiting current, related to the maximum rate at which a reactant can be supplied to an electrode [17]. Instead of directly introducing this parameter, a more specific approach develops material balances along each electrode thickness to evaluate the effective reactant and product composition at the electrode–electrolyte interface (TPB position) [1,18]. This formulation assumes that the electrochemical reaction occurs only at the layer boundary.

The present work develops a specific SOFC performance model by comparison between simulated and experimental data, to guarantee its physical validity. 0D stationary material balances are solved to predict global behaviour of an anode-supported solid oxide button cell. The proposed electrochemical kinetics is the optimization of a previously formulated simplified model, which assumes a linear dependence on ohmic and activation overpotential, while it neglects concentration contributions [9]. In the present work, a more complex formulation is considered. The ohmic and

activation terms are evaluated by a semi-empirical approach, where some variables are determined by laboratory tests. Meanwhile, the concentration overpotential is optimized solving 1D stationary material balances along electrode thickness to estimate the diffusion influence on the obtained voltage. It assumes that reactions occur in the electrode bulk volume. The least number of fitting parameters is used to avoid data overfitting and to obtain a more generic FC modelling.

## 2. Modelling

With reference to the experimental data of a solid oxide button cell, a 0D model is successfully used to simulate FC global performance supposing uniform distribution of temperature and pressure [17]. Considering the whole cell as control volume and assuming gas ideal behaviour, macroscale material balances are solved, at steady state, for each feeding component (Equation (4)). The generation term derives from Faraday theory.

$$N_{i,in} - N_{i,out} + \frac{v_i J}{zF} = 0 \tag{4}$$

So, in the electrochemical kinetics, fixed feeding temperature and pressure are considered, whereas the composition of components is an average value between inlet and outlet molar fractions, to take into account the concentration gradient on the cell plane [19]. According to the occurring global reaction (Equation (3)), the simulation considers a pure $H_2/N_2/H_2O$ mixture as anodic fuel and air as cathodic fuel; so present $N_2$ influences only diffusive transport mechanisms, not electrochemical processes.

### 2.1. Electrochemical Kinetics

From a thermodynamic point of view, the equilibrium voltage $E_{eq}$ is obtained by Nernst equation (Equation (5)), which represents the maximum performance of the fuel cell [20]:

$$E_{eq} = E^0 + \frac{RT}{zF} \ln \frac{p_{H_2,an} p_{O_2,cat}^{0.5}}{p_{H_2O,an}}, \tag{5}$$

where the reversible voltage $E^0$ derives from Gibbs free energy variation. In fact, considering Equations (6) and (7),

$$dG = -SdT + Vdp, \tag{6}$$

$$dG = -zFdE. \tag{7}$$

E dependence from temperature at constant pressure is identified according to Equation (8):

$$\left(\frac{dE}{dT}\right)_p = \frac{dS}{zF}. \tag{8}$$

Integrating between actual and standard values, Equations (9) and (10) are obtained [20]:

$$E^0 = E_{str} + \frac{\Delta S}{zF}(T - T_{str}), \tag{9}$$

$$E^0 = 1.253 - 2.4516 \cdot 10^{-4} T. \tag{10}$$

As mentioned in the introduction, under current load, the operating cell voltage is penalized by different irreversible losses (Equation (11)), known as overpotentials or polarization effects, which are function of operating conditions, cell design and materials.

$$V = E_{eq} - \eta_{ohm} - \eta_{act,an} - \eta_{act,cat} - \eta_{conc,an} - \eta_{conc,cat} \tag{11}$$

In the following, details are provided for each overpotential contribution.

### 2.1.1. Ohmic Overpotential

The ohmic overpotential $\eta_{ohm}$, expressed by Equation (12), is the result of both material resistances to current transfer and contact losses, namely resistances at interface between different cell layers.

$$\eta_{ohm} = R_{ohm}J + R_{cont}J \tag{12}$$

Neglecting contact losses and assuming a thermally activated charge transport mechanism for both ionic and electronic conduction [9], Equation (12) can be simplified in Equation (13):

$$\eta_{ohm} = R_{ohm}J = \frac{dT\exp^{\frac{E_{act,ohm}}{RT}}}{\sigma}J = P_1\exp^{\frac{P_2}{T}}J. \tag{13}$$

### 2.1.2. Activation Overpotential

The Butler–Volmer equation [20] describes the electrode polarization, considering the reversible redox reaction occurring at each semi-electrochemical cell (Equation (14)).

$$Ox + ze^- \leftrightarrow Red \tag{14}$$

The electrochemical rate (Equation (15)), a function of current density J through Faraday law, considers both the direct and indirect process (Equation (16)). It is assumed the dependency only from the composition of reactants.

$$v = k\prod_{i=1}^{n} a_{i,red}^{a_i} - k'\prod_{i=1}^{n} a_{i,ox}^{b_i} \tag{15}$$

$$J = zFk\prod_{i=1}^{n} a_{i,red}^{a_i} - zFk'\prod_{i=1}^{n} a_{i,ox}^{b_i} \tag{16}$$

Kinetic constants are expressed using an Arrhenius dependence, which considers contributions due to chemical and electrochemical phenomena owing to electrode polarization (Equation (17)):

$$J = zFk_{an,0}\exp^{\left(\frac{-E_{act,el}}{RT}\right)}\exp^{\left(\frac{\alpha z(V-E^0)F}{RT}\right)}\prod_{i=1}^{n} a_{i,red}^{a_i} - zFk_{cat,0}\exp^{\left(\frac{-E'_{act,el}}{RT}\right)}\exp^{\left(\frac{-\alpha'z(V-E^0)F}{RT}\right)}\prod_{i=1}^{n} a_{i,ox}^{b_i}. \tag{17}$$

The activity is substituted with the concentration of oxidized or reduced species at the electrode–electrolyte interface (TPB value). Lumping chemical process terms in the kinetic constants $k_{an}$ and $k_{cat}$, Equation (18) is obtained:

$$J = k_{an}\exp^{\left(\frac{\alpha z(V-E^0)F}{RT}\right)}\prod_{i=1}^{n} c_{i,red\_TPB}^{a_i} - k_{cat}\exp^{\left(\frac{-\alpha'z(V-E^0)F}{RT}\right)}\prod_{i=1}^{n} c_{i,ox\_TPB}^{b_i}. \tag{18}$$

Defining the electrode overpotential $\eta_{el}$ as the difference between actual voltage V and equilibrium voltage $E_{eq}$, Equation (18) is written as Equation (19):

$$J = k_{an}\exp^{\left(\frac{\alpha z(E_{eq}-E^0)F}{RT}\right)}\exp^{\left(\frac{\alpha z\eta_{el}F}{RT}\right)}\prod_{i=1}^{n} c_{i,red\_TPB}^{a_i} - k_{cat}\exp^{\left(\frac{-\alpha'z(E_{eq}-E^0)F}{RT}\right)}\exp^{\left(\frac{-\alpha'z\eta_{el}F}{RT}\right)}\prod_{i=1}^{n} c_{i,ox\_TPB}^{b_i}. \tag{19}$$

Multiplying bulk concentration, Equation (20) is obtained:

$$J = k_{an}\exp^{\left(\frac{\alpha z(E_{eq}-E^0)F}{RT}\right)}\exp^{\left(\frac{\alpha z\eta_{el}F}{RT}\right)}c_{i,red\_bulk}^{a_i}\prod_{i=1}^{n} \frac{c_{i,red\_TPB}^{a_i}}{c_{i,red\_bulk}^{a_i}} - k_{cat}\exp^{\left(\frac{-\alpha'z(E_{eq}-E^0)F}{RT}\right)}\exp^{\left(\frac{-\alpha'z\eta_{el}F}{RT}\right)}c_{i,ox\_bulk}^{b_i}\prod_{i=1}^{n} \frac{c_{i,ox\_TPB}^{b_i}}{c_{i,ox\_bulk}^{b_i}}. \tag{20}$$

As bulk composition is equal to TPB one at open circuit voltage (OCV), Equation (20) is simplified as follows (Equation (21)):

$$J = J_{0,el}\exp^{\left(\frac{\alpha z\eta_{el}F}{RT}\right)}\prod_{i=1}^{n} \frac{c_{i,red\_TPB}^{a_i}}{c_{i,red\_bulk}^{a_i}} - J'_{0,el}\exp^{\left(\frac{-\alpha'z\eta_{el}F}{RT}\right)}\prod_{i=1}^{n} \frac{c_{i,ox\_TPB}^{b_i}}{c_{i,ox\_bulk}^{b_i}}, \tag{21}$$

where $J_0$ is the exchange current density, an index of electrode material efficient as electrocatalyst; it represents the forward and reverse electrode reaction rate at the equilibrium state (OCV). In this

condition, the direct and inverse rates are equal, so the common Butler–Volmer equation is obtained (Equation (22)):

$$J = J_{0,el}\left[\exp\left(\frac{\alpha z \eta_{el} F}{RT}\right)\prod_{i=1}^{n}\frac{c_{i,red\_TPB}^{a_i}}{c_{i,red\_bulk}^{a_i}} - \exp\left(-\frac{\alpha' z \eta_{el} F}{RT}\right)\prod_{i=1}^{n}\frac{c_{i,ox\_TPB}^{b_i}}{c_{i,ox\_bulk}^{b_i}}\right].$$ (22)

The electrode overpotential in Equation (22) considers both activation and concentration effects, which become the main contributions at different conditions. Indeed, resistances due to reaction development are relevant at a low current, while component diffusion mechanisms are the rate-limiting step under a high load. The charge transfer coefficients are usually assumed to be 0.5 (equal requested energy for forward and backward process) because electrode material is a good catalyst for both reactions [14]. Neglecting the concentration gradient between bulk and TPB at a low load, the activation overpotential (Equation (23)) is written according to the hyperbolic sine form [13]:

$$\eta_{act,el} = \frac{RT}{\alpha z F}\sinh^{-1}\left(\frac{J}{2J_{0,el}}\right).$$ (23)

In Equation (23), the exchange current density $J_0$ is the unknown parameter, but it can be derived from Equations (20)–(22), considering just one oxidized and reduced compound (Equation (24)):

$$J_{0,el} = k_{an}c_{red\_bulk}^{a}\exp\left(\frac{z(E_{eq}-E^0)F}{2RT}\right) = k_{cat}c_{ox\_bulk}^{b}\exp\left(-\frac{z(E_{eq}-E^0)F}{2RT}\right).$$ (24)

An expression of $J_0$ is obtained from equivalence (Equation (24)), writing equilibrium constant to link direct and inverse kinetic terms (Equation (25)).

$$J_{0,el}^2 = k_{an}k_{cat}c_{red\_bulk}^{a}c_{ox\_bulk}^{b} = \frac{k_{an}^2}{k_{eq}}c_{red\_bulk}^{a}c_{ox\_bulk}^{b}$$ (25)

Equation (25) is rearranged by reintroducing Arrhenius dependence (Equation (17)) and substituting partial pressure working with gas as reactants and products. So $J_0$ formulation is obtained (Equation (26)):

$$J_{0,el} = \gamma_{el}p_{red\_bulk}^{\frac{a}{2}}p_{ox\_bulk}^{\frac{b}{2}}\exp^{-\frac{E_{act,el}}{RT}}.$$ (26)

Hence, anodic and cathodic current densities (Equations (27) and (28)) are determined by a power law expression, regarding the dependency on gas component composition, multiplied by an Arrhenius-type term to consider the influence of temperature [15].

$$J_{0,an} = P_3\left(\frac{p_{H_2\_bulk}}{p_{str}}\right)^A\left(\frac{p_{H_2O\_bulk}}{p_{str}}\right)^B\exp^{-\frac{E_{act,an}}{RT}}$$ (27)

$$J_{0,cat} = P_4\left(\frac{p_{O_2\_bulk}}{p_{str}}\right)^C\exp^{-\frac{E_{act,cat}}{RT}}$$ (28)

The reaction orders A and B are usually defined experimentally. Conversely, C can be determined from theoretical approach, as the cathodic rate-determining step is well defined as the oxygen ion formation and incorporation into the electrolyte [21]:

$$O_\sigma + V_0^{..}(YSZ) + 2e^- \leftrightarrow O_\sigma^x(YSZ) + \sigma,$$ (29)

where $O_\sigma$ is the oxygen adsorbed at cathode, $\sigma$ the vacancy inside electrode, $V_0^{..}$ the oxygen vacancy within YSZ electrolyte, and $O_\sigma^x$ the oxygen in YSZ lattice according to Kronger–Vink notation.

Equation (25) is applied for the described electrochemical process (Equation (29)), considering component activities. Specifically, $V_0^{..}$ and $O_\sigma^x$ activities are constant, depending only on electrolyte composition, so they are neglected; reaction orders are assumed equal to stoichiometric coefficients (Equation (30)).

$$J_{0,cat}^2 = k_{an}k_{cat}a_{O_\sigma}a_\sigma$$ (30)

According to a thermodynamic approach, Equation (31) is valid at equilibrium condition:

$$\sum_{i=1}^{n} \mu_i \nu_i = 0, \tag{31}$$

where the chemical potential $\mu$ is defined by Denbigh [22], according to Equation (32):

$$\mu_i = \mu_i^0 + RT\ln a_i. \tag{32}$$

Because the oxygen adsorption is an equilibrium reaction (Equation (33)),

$$O_2 + 2\sigma \leftrightarrow 2O_\sigma, \tag{33}$$

Equation (31) is valid and so the followed correlation is written (Equation (34)):

$$\mu_{O_2} + 2\mu_\sigma = 2\mu_{O_\sigma}. \tag{34}$$

Substituting Equation (32) into Equation (34) and considering partial pressure for an ideal gas, Equation (35) is obtained:

$$\mu_{O_2}^0 + RT\ln p_{O_2} + 2\mu_\sigma^0 + 2RT\ln a_\sigma = 2\mu_{O_\sigma}^0 + 2RT\ln a_{O_\sigma}. \tag{35}$$

Solving Equation (35), Equation (36) comes out as follows:

$$\frac{a_{O_\sigma}}{a_\sigma} = \text{const } p_{O_2}^{0.5}. \tag{36}$$

Substituting Equation (36) into Equation (30), the correlation between $J_0$ and oxygen partial pressure is derived (Equation (37)):

$$J_{0,cat} \propto a_{O_\sigma} p_{O_2}^{0.25} = a_{O_\sigma} p_{O_2}^{C}. \tag{37}$$

The cathodic kinetic order C can be assumed to be equal to 0.25 [21].

### 2.1.3. Concentration Overpotential

The concentration overpotential derives from Butler–Volmer (Equation (22)), but, in this case, the electrochemical reaction is favoured and so the exchange current density tends to an infinite value. So, Equation (38) is derived:

$$\exp^{\left(\frac{\alpha z \eta_{conc,el} F}{RT}\right)} \prod_{i=1}^{n} \frac{c_{i,red\_TPB}^{a_i}}{c_{i,red\_bulk}^{a_i}} - \exp^{\left(-\frac{\alpha' z \eta_{conc,el} F}{RT}\right)} \prod_{i=1}^{n} \frac{c_{i,ox\_TPB}^{b_i}}{c_{i,ox\_bulk}^{b_i}} = 0. \tag{38}$$

The concentration overpotential results to be the following (Equation (39)):

$$\eta_{conc,el} = \frac{RT}{2\alpha z F} \ln\left( \prod_{i=1}^{n} \frac{c_{i,red\_bulk}^{a_i} c_{i,ox\_TPB}^{b_i}}{c_{i,ox\_bulk}^{b_i} c_{i,red\_TPB}^{a_i}} \right). \tag{39}$$

This term considers the concentration gradient created under current load, when the transport of reactants and products from and to electrochemical reaction sites is too slow to maintain initial bulk compositions. The requirement of reactants exceeds the gas capability to diffuse through porous materials. Consequently, there is an undersupply of fuel at anode (or oxidant at cathode); simultaneously, the produced water is transported out of the TPB sites too slowly. The cathodic concentration gradient is usually relevant only at low oxygen partial pressure ($p_{O2} < 0.05$ atm) [15], so it is neglected when air is used. Meanwhile, considering involved reactants and products, substituting gas partial pressures and assuming $\alpha$ equal to 0.5, the anodic contribution is obtained (Equation (40)):

$$\eta_{conc,an} = \frac{RT}{zF} \ln\left( \frac{p_{H_2\_bulk}^{a} p_{H_2O\_TPB}^{b}}{p_{H_2O\_bulk}^{b} p_{H_2\_TPB}^{a}} \right). \tag{40}$$

In $\eta_{conc,an}$, gas partial pressures at TPB can be determined solving material balances along the anode thickness. Gas motion is the result of different mechanisms: convection, diffusion, and induced

convection if an asymmetric system is present. As the pressure gradient is insignificant inside pores, the first term is not considered. Occurring an equimolar counter-current diffusion of reactants and products at anode, there are not induced convection flows. Thereby, the transport is only the result of diffusion, which is described by Fick theory, the simplest and most common approach for gas motion inside porous media [23]. The anode material is a cermet (NiYSZ), which can conduct both ions and electrons, so the reaction occurs in the whole electrode volume. According to these hypotheses, the material balance is expressed by Equation (41), taking account of specific boundary conditions set at the extremities of the anode (with thickness $d_{an}$) as in Equations (42) and (43).

$$\frac{D_i^{eff}}{RT}\frac{d^2p_i}{dx^2} = \pm\frac{J}{zFd} \tag{41}$$

$$p_i(x = 0) = p_{i\_bulk} \tag{42}$$

$$-\frac{D_i^{eff}}{RT}\frac{dp_i}{dx}|_{x=d_{an}} = 0 \tag{43}$$

Specifically, Equation (42) assumes a homogeneous distribution of reactants at the interconnect-anode limit. Equation (43) is justified by the fact that no flow can pass at interface using a dense electrolyte to avoid cross-over. Solving Equation (41), $H_2$ and $H_2O$ partial pressure profiles along the anodic layer are calculated, as resulting from Equations (44) and (45), respectively.

$$p_{H_2} = p_{H_2\_bulk} + \frac{JRTx^2}{2zFd_{an}D_{H_2}^{eff}} - \frac{JRTx}{zFD_{H_2}^{eff}} \tag{44}$$

$$p_{H_2O} = p_{H_2O\_bulk} - \frac{JRTx^2}{2zFd_{an}D_{H_2O}^{eff}} + \frac{JRTx}{zFD_{H_2O}^{eff}} \tag{45}$$

As the reaction occurs in the whole electrode volume, TPB partial pressures are not evaluated distinctively. A good approximation consists of an average value calculated along the anode profile (Equation (46)).

$$p_{i\_TPB} = \frac{\int_0^{d_{an}} p_i dx}{d_{an}} \tag{46}$$

Substituting Equations (44) and (45) into Equation (46), the average values of TPB partial pressure are calculated (Equations (47) and (48)):

$$p_{H_2\_TPB} = p_{H_2\_bulk} - \frac{JRTd_{an}}{3zFD_{H_2}^{eff}}, \tag{47}$$

$$p_{H_2O\_TPB} = p_{H_2O\_bulk} + \frac{JRTd_{an}}{3zFD_{H_2O}^{eff}}. \tag{48}$$

The coefficient $D_i^{eff}$ is a function of molecular diffusivity, but it is correct considering specific electrode porosity $\varepsilon$ and tortuosity $\xi$ (Equation (49)).

$$D_i^{eff} = \frac{\varepsilon}{\xi}D_{i,mix} \tag{49}$$

In Equation (49), the molecular diffusivity is taken into account for a gas binary mixture on the basis of diffusion coefficient $D_{i-j}$ (Equation (50)), weighted on molar fractions (Equation (51)) [24]. Specifically, the molecular diffusion coefficient $D_{i-j}$ is calculated using the Fuller approach for binary system [14]:

$$D_{i-j} = \frac{0.00143T^{1.75}}{p(\frac{2}{\frac{1}{M_i} + \frac{1}{M_j}})^{0.5}(v_i^{\frac{1}{3}} + v_j^{\frac{1}{3}})^2}, \tag{50}$$

$$D_{i,mix} = \sum_{i \neq j} \left( \frac{y_j}{D_{i-j}} \right)^{-1} (1 - y_i). \tag{51}$$

Thus, $\eta_{conc,an}$ is calculated substituting Equations (47) and (48) into Equation (40), as shown in the following (Equation (52)):

$$\eta_{conc,an} = \frac{RT}{zF} \ln \left( \frac{(1 + \frac{RTd_{an}J}{3zFD_{H_2O}^{eff} p_{H_2O\_bulk}})^b}{(1 - \frac{RTd_{an}J}{3zFD_{H_2}^{eff} p_{H_2\_bulk}})^a} \right). \tag{52}$$

In summary, referring to the previous Equation (11), the cell potential is evaluated by means of Equation (53).

$$V = E_{eq} - P_1 T \exp^{\frac{P_2}{T}} J - \frac{RT}{F} \sinh^{-1} \left( \frac{J}{2J_{0,an}} \right) - \frac{RT}{2F} \sinh^{-1} \left( \frac{J}{2J_{0,cat}} \right) - \frac{RT}{2F} \ln \left( \frac{(1 + \frac{RTd_{an}J}{6FD_{H_2O}^{eff} p_{H_2O\_bulk}})^b}{(1 - \frac{RTd_{an}J}{6FD_{H_2}^{eff} p_{H_2\_bulk}})^a} \right) \tag{53}$$

## 3. Design of Experiments

A specific experimental campaign is performed to study how different operating parameters influence cell performance. Every variable is changed one at a time to isolate single effects. In each test, the anodic flow rate is humidified to guarantee a good operation (about 3%–4% $H_2O$). $H_2$ is diluted with $N_2$ at three temperatures (1023–1048–1073 K), when thermal activated processes are favoured. To underline the fuel composition influence on cell performance, three different hydrogen concentrations are tested (100%–50%–25% $H_2$ on a volume basis). The tests are also performed changing anodic and cathodic flow rates. All test parameters are resumed in Table 1.

**Table 1.** Tests at different fuel molar compositions, temperatures, anodic, and cathodic flow rates.

| T [K] | Anode | | | Cathode | | |
|---|---|---|---|---|---|---|
| | Flow Rate [mL/min] | %$H_2$ | %$N_2$ | Flow Rate [mL/min] | %$O_2$ | %$N_2$ |
| 1023 | 100–150–200 | 100 | - | 200–300–400 | 21 | 79 |
| 1023 | 100–150–200 | 50 | 50 | 200–300–400 | 21 | 79 |
| 1023 | 100–150–200 | 25 | 75 | 200–300–400 | 21 | 79 |
| 1048 | 100–150–200 | 100 | - | 200–300–400 | 21 | 79 |
| 1048 | 100–150–200 | 50 | 50 | 200–300–400 | 21 | 79 |
| 1048 | 100–150–200 | 25 | 75 | 200–300–400 | 21 | 79 |
| 1073 | 100–150–200 | 100 | - | 200–300–400 | 21 | 79 |
| 1073 | 100–150–200 | 50 | 50 | 200–300–400 | 21 | 79 |
| 1073 | 100–150–200 | 25 | 75 | 200–300–400 | 21 | 79 |

For each test, both i-V curves and electrochemical impedance spectroscopies (EISs) are measured. EIS permits the internal resistance identification in OCV for every considered temperature.

## 4. Results and Discussion

The model parameters are identified and, later, the simulation is validated by comparison with experimental data.

### 4.1. Identification of Parameters

The OCV conditions are evaluated through Nernst formulation (Equation (5)), nevertheless, theoretical values were higher than measured ones. The causes are the gaseous or/and electronic

leakages; some electrons can be conducted through the electrolyte, but a defective sealing is usually the main cause of this gap [19,25]. For this reason, an additional overpotential term of 0.03 V is introduced to consider this phenomenon (Equation (54)).

$$E_{eq} = E^0 + \frac{RT}{zF} \ln \frac{p_{H_2,an} p_{O_2,cat}^{0.5}}{p_{H_2O,an}} - \eta_{leakage} \tag{54}$$

The ohmic overpotential is a function only of temperature (Equation (13)). $P_1$ and $P_2$ are evaluated fitting cell internal resistances, provided by EIS analysis, at different temperatures. Plotting in logarithmic scale, as expected, a linear dependence is observed (Figure 1).

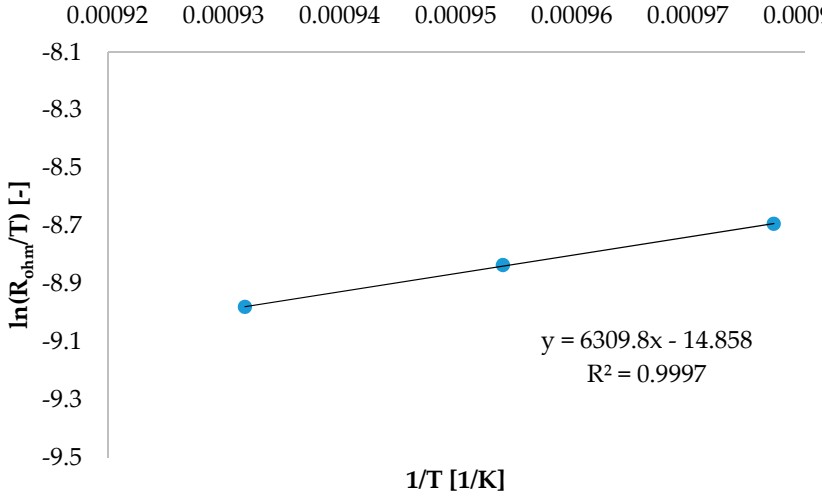

**Figure 1.** Temperature dependence of internal resistance: experimental data and fitting.

The obtained $P_1$-$P_2$ pair is used for fuel cell simulation (Table 2).

**Table 2.** Ohmic resistance parameters.

| Parameter | Used Value |
|---|---|
| $P_1$ [$\Omega m^2/K$] | $3.53 \cdot \times 10^{-11}$ |
| $P_2$ [K] | 6309.80 |

The activation overpotential, corresponding to the energy requested to allow electrochemical reaction occurrence, is considered a thermal activated process, using an Arrhenius-type equation (Equations (23), (27), and (28)). Activation energy values are well defined in the literature for both anodic and cathodic sides (Table 3).

**Table 3.** Anodic and cathodic activation energy.

| Parameter | Used Value | Literature Values |
|---|---|---|
| $E_{act,an}$ [kJ/mol] | 110 | 100–120 [17] |
| $E_{act,cat}$ [kJ/mol] | 120 | 110–160 [17] |

Both anodic and cathodic processes are multi-step mechanisms, which include adsorption–desorption, diffusion, and charge transfer [16]. Yet, the kinetic rate is influenced, above all, from the slowest phenomenon. For the anodic side, different reaction orders are proposed, as the rate-limiting step is not detected in a univocal way. In some cases, these values also change as a function of the operating condition, such as $H_2$ and $H_2O$ partial pressure [25]. The present work proposes the

followed values for A and B (Table 4). Meanwhile, the cathodic reaction order is equal to 0.25, as confirmed by both the theoretical and experimental point of view [21].

**Table 4.** Anodic reaction orders.

| Parameter | Used Value | Literature Values |
|-----------|-----------|-------------------|
| A [–] | 0.50 | 0.1–2 [26] |
| B [–] | 0.55 | −0.5–1 [26] |

Finally, the pre-exponential terms $P_3$ and $P_4$ depend on electrode material and its properties as electrocatalysts. They are specific for each cell, so they are calculated fitting the experimental i-V curves at three anodic fuel compositions, to evaluate $H_2$ partial pressure effects and to tune the model (Figure 2).

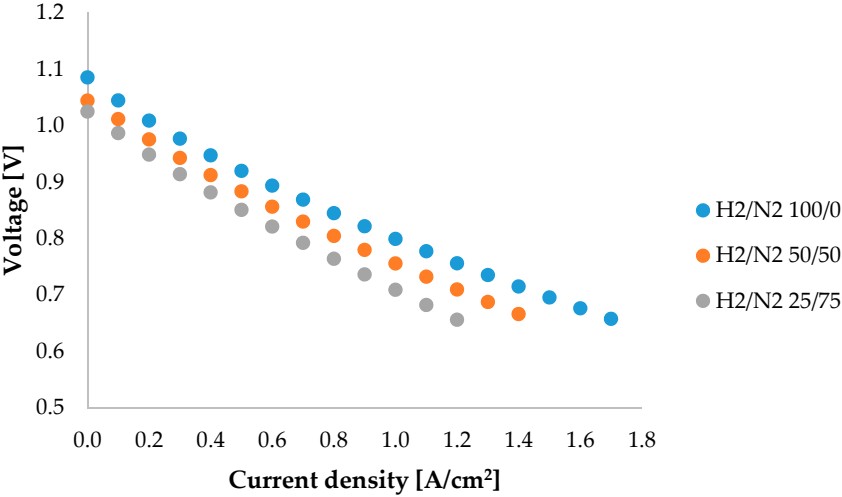

**Figure 2.** Experimental i-V curves at 1048 K (150 mL/min anodic and 300 mL/min cathodic flow rate).

The obtained fitting i-V curves show a good agreement with experimental data (Figure 3).

(a)

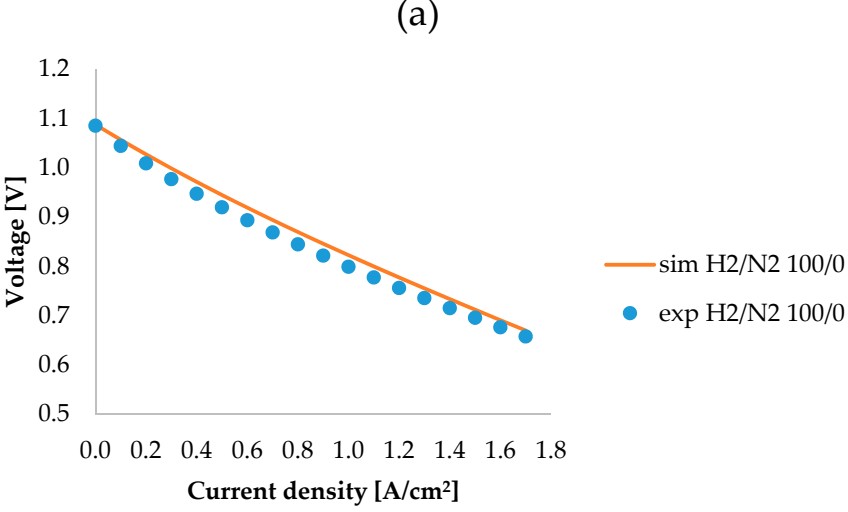

(b)

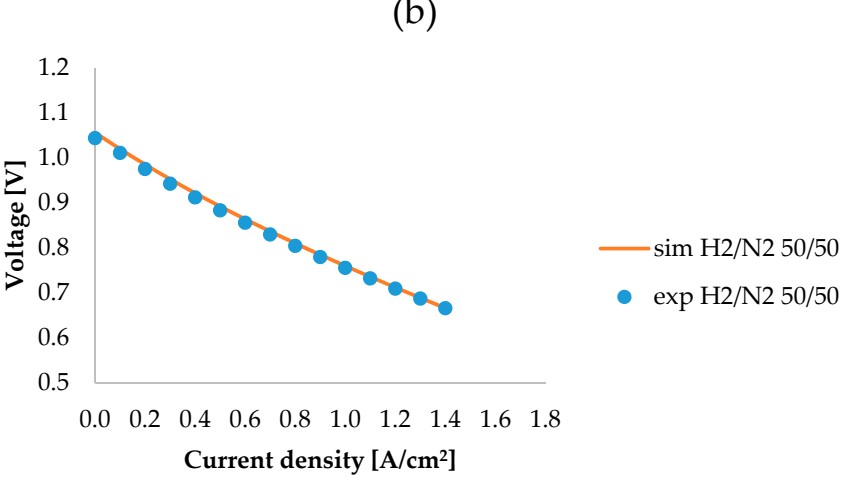

(c)

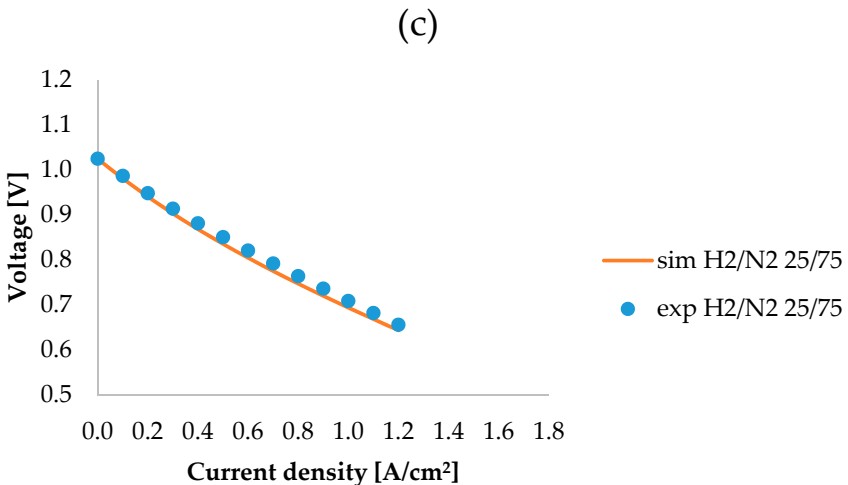

**Figure 3.** Fitting of i-V curves at 1048 K: (**a**) $H_2/N_2$ 100/0, (**b**) $H_2/N_2$ 50/50, and (**c**) $H_2/N_2$ 25/75.

The obtained fitting parameters $P_3$ and $P_4$ are comparable to literature values (Table 5).

**Table 5.** Anodic and cathodic pre-exponential factors.

| Parameter | Used Value | Literature Values |
|---|---|---|
| $P_3$ [A/m²] | $2.8 \cdot \times 10^9$ | $5.5 \cdot \times 10^8$–$5.5 \cdot \times 10^{10}$ [15] |
| $P_4$ [A/m²] | $4 \times 10^{10}$ | $7.0 \cdot \times 10^8$–$7.0 \cdot \times 10^{10}$ [15] |

The concentration overpotential estimates the resistance as a result of gas diffusion along the electrode to reach TPB active sites. In this term, the unknown parameters are effective diffusion coefficients $D_i^{eff}$, which are calculated using the Fuller approach, whose parameters are retrieved in the literature (Table 6).

**Table 6.** Effective diffusion coefficient parameters.

| Parameter | Used Value | Reference |
|---|---|---|
| $v_{H2}$ [−] | 6.12 | [27] |
| $v_{H2O}$ [−] | 13.10 | [27] |
| $v_{N2}$ [−] | 18.50 | [27] |
| $\varepsilon$ [−] | 0.30 | [5] |
| $\xi$ [−] | 4.00 | [2] |

The obtained effective diffusion coefficients for model tuning are represented in Table 7.

**Table 7.** Effective diffusion coefficients.

| Parameter | $H_2/N_2$ 100/0 | $H_2/N_2$ 50/50 | $H_2/N_2$ 25/75 |
|---|---|---|---|
| $D_{H2}^{eff}$ [m²/s] | 6.23·× 10⁻⁵ | 5.40·× 10⁻⁵ | 5.38·× 10⁻⁵ |
| $D_{H2O}^{eff}$ [m²/s] | 6.23·× 10⁻⁵ | 2.75·× 10⁻⁵ | 2.15·× 10⁻⁵ |

*4.2. Preliminary Model Validation*

After using some of the experimental results for model tuning and parameter identification, other data are useful to check the simulation. Specifically, the model validity is confirmed by comparing tests performed at temperatures lower and higher than the values used for model tuning. The proposed SOFC simulation shows a good agreement with experimental characteristic curves (Figures 4 and 5). The relative error is calculated considering the difference between measured and simulated data at several current loads. In all cases, the difference is always lower than 3%. Therefore, the assumptions undertaken have allowed for successful performance modelling in spite of the pseudo-macroscale approach.

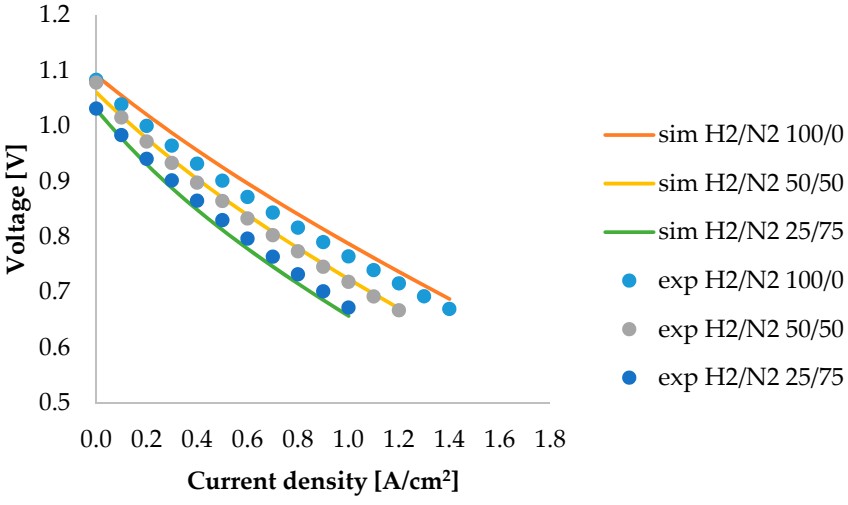

**Figure 4.** Experimental and simulated i-V curves at 1023 K at different fuel compositions.

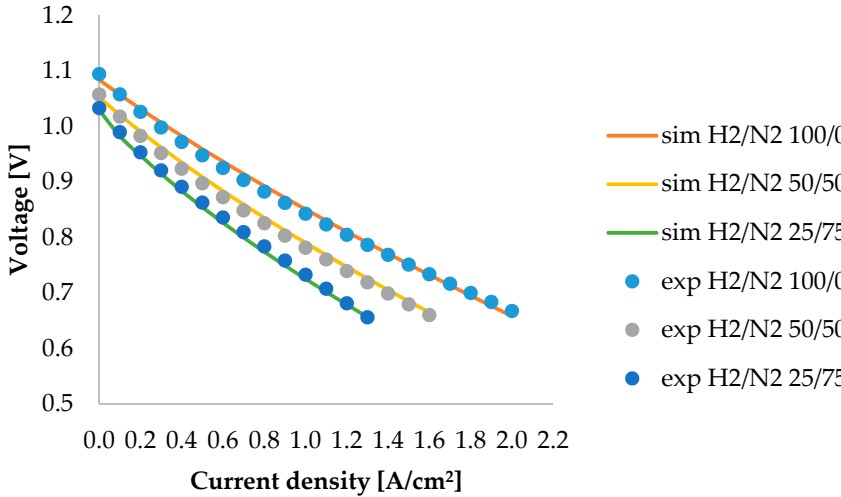

**Figure 5.** Experimental and simulated i-V curves at 1073 K at different fuel compositions.

As expected, temperature rise worsens the Nernst potential (an exothermic phenomenon occurs). However, global performance improves as a result of polarization reduction. $H_2$ composition increase also favours the process, as the results show.

A major accuracy is obtained in comparison with previously developed code, where linear dependences were supposed [9]. The hyperbolic sine activation overpotential equation allows a better simulation of the characteristic curve profile. In agreement with the literature, the linear model is valid only in specific operating conditions [13]. The main difference is seen at low $H_2$ partial pressure, where a more specific formulation is requested to evaluate the relevant fuel composition influence (Figure 6).

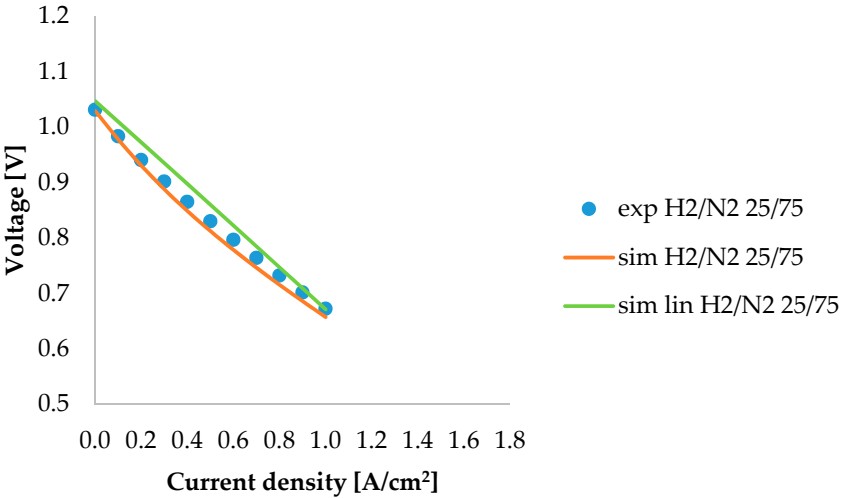

**Figure 6.** Comparison between experimental data and different simulated models at 1023 K (present model and previous linear one).

The i-V curve simulation is a useful tool to forecast cell behaviour, but a better optimization is obtained knowing specifically single polarization terms. Each contribution has a different relevance, changing the current density (Figure 7). At a low load, the activation overpotential has a big influence, as well as the ohmic one. When current increases, the ohmic overpotential becomes the main resistance because the activation terms reach an asymptotic value. As discussed in the literature [1],

the concentration contribution provides the lowest effect, until the system works at a current load less than the limiting current density (not detected in the considered range of experimental conditions).

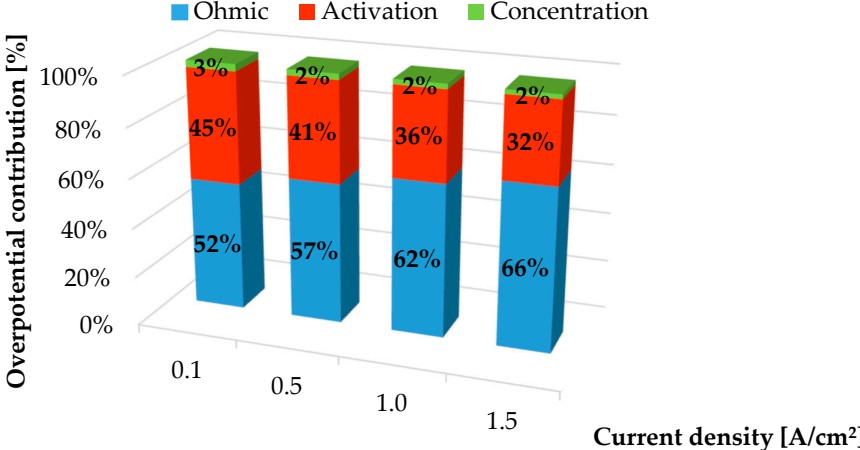

**Figure 7.** Calculated overpotential terms changing the current load at 1048 K and $H_2/N_2$ 100/0.

The polarization contribution analysis permits to underline dependences between operating conditions and electrochemical kinetics. Thus, the rate-limiting step is detected and process optimization can be made. Temperature influences all overpotentials because they are thermal activated processes (Figure 8). If temperature rises, the ohmic term decreases owing to faster ions conduction. The anodic and cathodic electrochemical reactions are favoured as well, requesting a small activation energy. According to simulation, anodic concentration overpotential has a lower thermal dependence [28]. The diffusion coefficient improves at a high temperature, yet gas density decreases as temperature rises, so profiles do not show relevant differences owing to the overlap of these two effects [29].

(a)

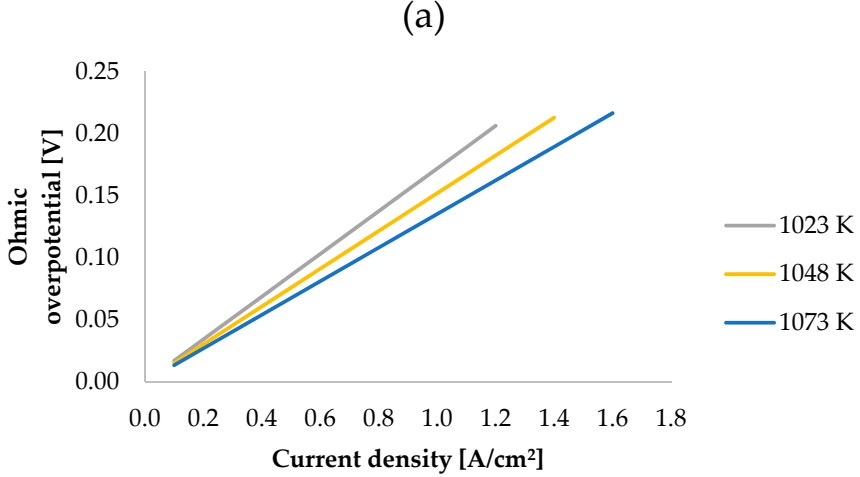

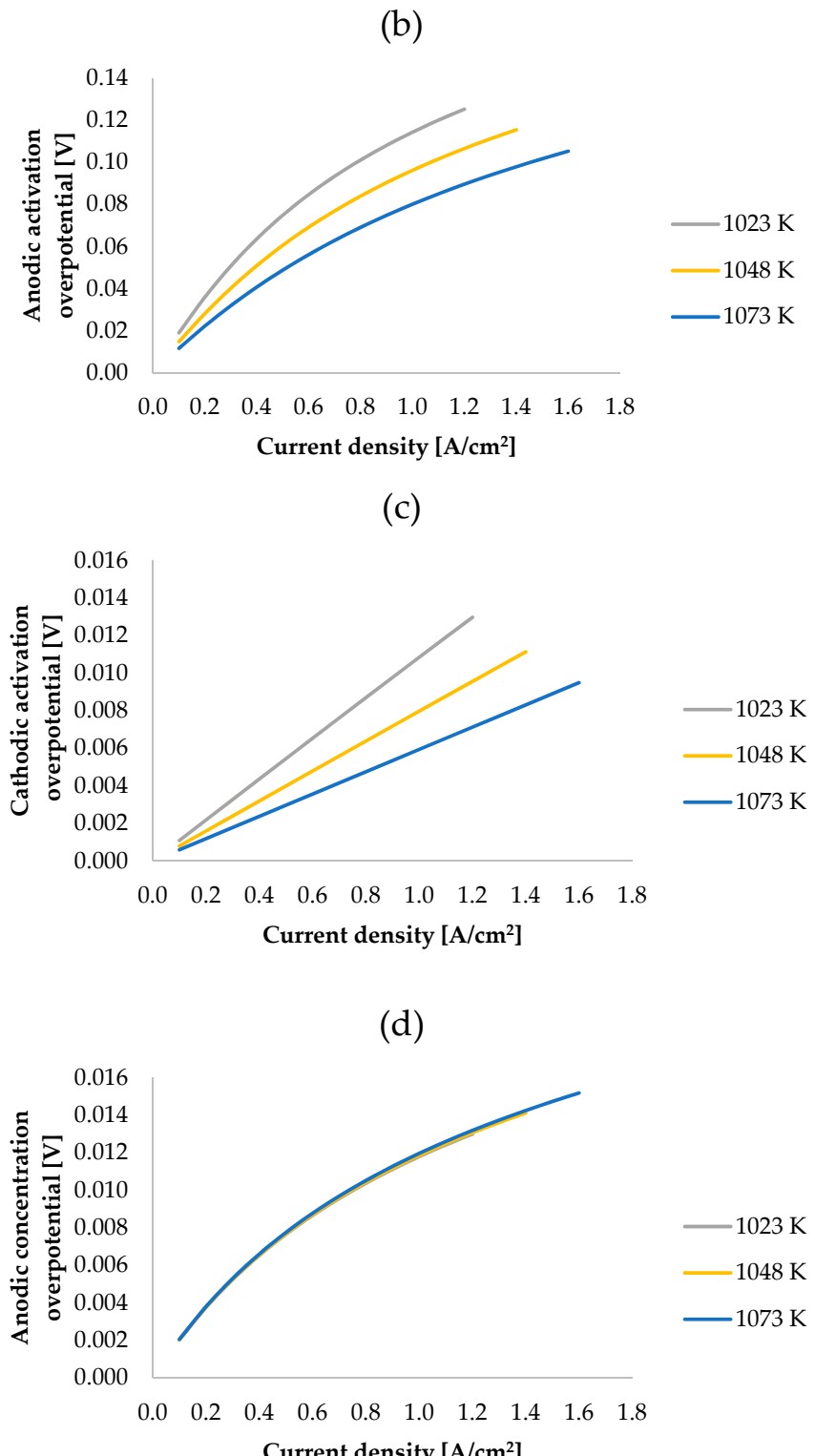

**Figure 8.** (**a**) Ohmic, (**b**) anodic activation, (**c**) cathodic activation, and (**d**) anodic concentration overpotential calculated at $H_2/N_2$ 50/50 and different temperatures.

Also, the anodic gas composition affects the polarization contribution. In particular, the studied conditions focus on fuel concentration. Higher $H_2$ concentration favours the electrochemical reaction, so both the anodic activation term and material diffusion mechanism improve. Obviously, no correlations are seen with the ohmic overpotential, as well as with the cathodic activation one (Figure 9).

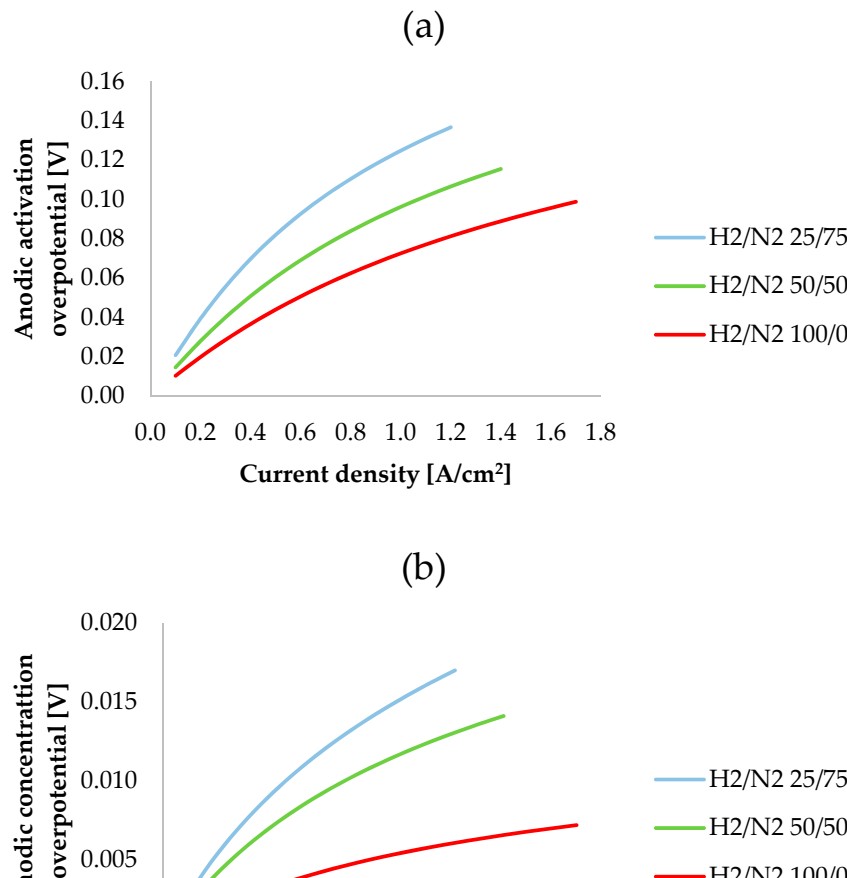

**Figure 9.** (**a**) Anodic activation, and (**b**) anodic concentration overpotential calculated at 1048 K and different $H_2$ compositions.

Finally, laboratory tests are carried out at different anodic and cathodic flow rates. No influences are underlined from both an experimental and a theoretical point of view.

## 5. Materials and Methods

Experiments were performed on a planar NiYSZ/8YSZ/GDC-LSCF button cell. The main structural support consisted of the anode layer (NiYSZ), whose outer diameter and thickness were 28 mm and 240 μm, respectively. The electrolyte (8YSZ) was sintered onto the anode and was 8 μm thick, while a 50 μm-thick cathode layer (GDC-LSCF) was screen-printed over the electrolyte–anode assembly. The surface assumed as the cell active area is the smallest among cell electrodes and metal current collectors (1 cm²). The button cell was sealed onto a dense alumina housing with a high temperature glass paste (Schott G018-311), exhibiting a thermal expansion coefficient compatible

with $ZrO_2$. The sealing paste was cured according to instructions recommended by the supplier, anyhow without exceeding a temperature rate of 1 K/min (Figure 10).

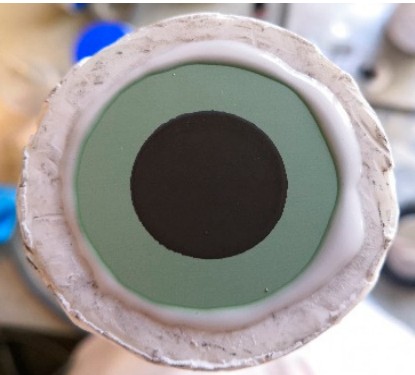

**Figure 10.** Button cell sealed on the alumina support.

The cell housing was set into an electric furnace, where two K-thermocouples allowed temperature regulation and measurement (for the complete description of the cell housing geometry and test bench, refer to the work of [30]). Technical-purity $H_2$ and $N_2$ were supplied to the anode, regulating gas supply with digital mass flow meter controllers (Vögtlin Red-y Smart, accuracy of 0.2% on the full scale). The anode feeding gas was humidified with a water bubbler kept at controlled temperature, hence achieving a moisture concentration of 3%–4% vol. Button cell electrodes were electrically accessed by a 99.999% purity gold mesh (cathode) and a 99.999% purity nickel mesh (anode). No reference electrode was present.

The cell start-up was based on a standard procedure, to reach a temperature of 800 °C. Then, the anode was reduced, introducing $H_2$ in feeding. After 50 h from the reduction completion, the cell was stable and ready for the electrochemical characterization.

All in-operando electric analyses were measured with a BioLogic SP-240 analyzer (Biologic, Seyssinet-Pariset, France). Voltage measurement range was set to the interval 0.5 V–1.5 V, so that sampling resolution was very high (20 μV). The current range was set to 4 A. The i-V curves were recorded with a potentiostatic method, applying a voltage ramp equal to −40 mV/min from OCV to 0.7 V, and then +40 mV/min from 0.7 V back to OCV. EIS spectra were sampled in galvanostatic mode with a single-sine method, supplying a 20 mA-amplitude current signal. The measurement was investigated from 200 kHz to 100 mHz, acquiring 10 points in each frequency decade. Between the consequent cycles, a wait phase of 2 min allowed performance stabilization, to prevent transients and artefacts on impedance measurements.

## 6. Conclusions

The SOFC electrochemical model is developed considering different polarization contributions, which penalize cell performance. The theoretical elaboration is paired with experimental activity to guarantee reliable results. Operating conditions, such as temperature, fuel composition, and flow rate, are changed once at time to underline specific effects on cell behaviour. The comparison with laboratory tests is used for both model parameter identification and subsequent code preliminary validation. The proposed modelling is based on 0D macroscale material balances, testing an anode-supported button cell. As the electrochemical kinetics is influenced by gaseous reactant and product transport, 1D local equations are also solved along the anodic electrode thickness. A non-linear correlation for the activation overpotential and a TPB composition correction for the concentration overpotential permit a better simulation of characteristic i-V curves, which have a linear profile at a low current and an increased slope as the load rises. A good agreement between simulated and tested data preliminarily demonstrates the validity of the proposed electrokinetic model, which could be successfully used as a baseline to develop further studies, like 2D or 3D cell modelling or the



extension to a wider range of operating conditions, such as that already performed for other fuel cell types [31,32].

**Author Contributions:** Conceptualization, F.R.B. and B.B.; Investigation, A.B. and L.B.; Methodology, F.R.B. and B.B.; Writing—original draft, F.R.B.; Writing—review & editing, F.R.B., B.B., A.B., and L.B.

**Funding:** The present research work has received funding from the Italian Ministry of Education, Universities and Research, MIUR, as Project of National Interest, PRIN 2017F4S2L3.

**Acknowledgments:** The authors wish thank Samuele Delfino for his contribution in the preliminary analysis of experimental data.

**Conflicts of Interest:** The authors declare no conflict of interest.

## Nomenclature

| | |
|---|---|
| A, B, C | Power law coefficients for exchange current density [–] |
| a | Activity |
| *a,b* | Kinetic orders [–] |
| c | Concentration [mol/m$^3$] |
| D | Diffusion coefficient [m$^2$/s] |
| d | Thickness [m] |
| E | Potential [V] |
| E$^0$ | Reversible voltage [V] |
| E$_{act}$ | Activation energy [J/mol] |
| F | Faraday constant [C/mol] |
| G | Gibbs free energy [J/mol] |
| J | Current density [A/m$^2$] |
| J$_0$ | Exchange current density [A/m$^2$] |
| k | Kinetic constant |
| k$_{eq}$ | Equilibrium constant |
| M | Molecular weight [mol/g] |
| N | Molar flux [mol/(m$^2$s)] |
| n | Number of components [–] |
| P | Electrochemical parameters |
| p | Pressure [atm] |
| R | Resistance [Ω] |
| *R* | Gas constant [J/(molK)] |
| S | Entropy [J/(molK)] |
| T | Temperature [K] |
| V | Cell voltage [V] |
| v | Diffusion volume [–] |
| *v* | Reaction rate |
| x | space [m] |
| y | Molar fraction [–] |
| z | Number of transferred electrons [–] |
| Greek letters | |
| $\alpha$ | Charge transfer coefficient [–] |
| $\gamma$ | Pre-exponential coefficient in exchange current density |
| $\varepsilon$ | Porosity [–] |
| $\mu$ | Chemical potential |
| $\eta$ | Overpotential [V] |
| $\nu$ | Stoichiometric coefficient [–] |
| $\xi$ | Tortuosity [–] |
| $\sigma$ | Conductivity [1/(Ωm)] |
| Subscript | |
| act | Activation |
| an | Anode |
| bulk | Bulk electrode |
| cat | Cathode |
| conc | Concentration |

| cont | Contact |
| eff | Effective |
| el | Electrode |
| eq | Equilibrium |
| in | Inlet |
| mix | Mixture |
| ohm | Ohmic |
| out | Outlet |
| ox | Oxidized species |
| red | Reduced species |
| str | Standard |
| TPB | Three phase boundary |

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
