# Peer review of "Optimization of a Reference Kinetic Model for Solid Oxide Fuel Cells"

_catalysts, doi:10.3390/catal10010104_

Round 1

Reviewer 1 Report

The ability to store energy from intermittent renewable resources is a significant challenge that can be addressed by generating hydrogen as a fuel via electrochemical water splitting. Fuel cell is an excellent device to generate electricity with no harmful chemical reactions but the voltage for single cell is low (when compared to that of battery). Usually catalysts for chemical reactions belong to the noble metal group and they are expensive. Recently, significant progress has been made in developing non-precious metal-based earth abundant catalysts for better OER performance. The presented kinetic approach for 0D system and its validation is significance to the submitted journal and may bring some useful insights to the community. Before rendering a final decision.

My specific points are below:

In section 2, section 2.1 (Electrochemical kinetics) is quite well known containing several equations until Eq. (53); all these can be moved to appendix. Purity of hydrogen and H2O need to be mentioned. Section 4.1 Little bit more reasoning is required to justify why theoretical OCV values were higher than the measured ones. The usual convention for resistance is (R) and for pressure is (P). Being this as the situation, why the terms Pi and P2 are used for cell internal resistances? Does the internal resistance refer to Solution resistance and Charge transfer resistance? If so, then it needs to be quoted as Rs and Rct, respectively. Tables 3 -5, third column (Reference) term is bit confusing. Should be stated as Theoretical value [from the literature]. Table 6 is OK. In Fig. 7; please include the units for overpotential. From Fig. 8 -9 what authors have inferred from the experimental tests and the obtained values of overpotential? Please comment. Section 5. When there is no reference electrode, then potentials are measured with respect to anode??? Please clarify. Section 5: Line 373 “Between an EIS scan and the following next” can be re-worded as “between the consequent cycles”

Reviewer 2 Report

To my judgment, this manuscript could provide some contribution to the field. However, there are still some details need to be paid attention to.

Page 2, line 64: The relationship between the ohmic resistance and the cell geometry is further emphasized by the following reference; doi.org/10.1016/j.ssi.2006.10.014

Eq. 29: The authors attribute the origin of the oxygen vacancy to the YSZ electrolyte. However, does the vacancy of the LSCF cathode contribute to?

Fig. 7: The impedance spectra recorded even under OCVs may support the simulated proportion of the ohmic and total activation overpotintials.

Methods: Which is the four- or two-probe method used for the electrochemical measurements?
